# The Diagnostic Performance of Various Clinical Specimens for the Detection of COVID-19: A Meta-Analysis of RT-PCR Studies

**DOI:** 10.3390/diagnostics13193057

**Published:** 2023-09-26

**Authors:** Khaled Sadeq Ali Al-Shaibari, Haider Abdul-Lateef Mousa, Mohammed Abdullah A. Alqumber, Khaled A. Alqfail, AbdulHakim Mohammed, Khalid Bzeizi

**Affiliations:** 1College of Medicine, Najran University, Najran 11001, Saudi Arabia; 2College of Medicine, University of Basrah, Basrah 64001, Iraq; 3Laboratory Medicine Department, Albaha University, Albahah 65731, Saudi Arabia; 4Department of Liver Transplantation, King Faisal Specialist Hospital and Research Center, Riyadh 13541, Saudi Arabia

**Keywords:** COVID-19, diagnosis, sensitivity, specificity

## Abstract

Background: The diagnostic performance of numerous clinical specimens to diagnose COVID-19 through RT-PCR techniques is very important, and the test result outcome is still unclear. This review aimed to analyze the diagnostic performance of clinical samples for COVID-19 detection by RT-PCR through a systematic literature review process. Methodology: A compressive literature search was performed in PubMed/Medline, Scopus, Embase, and Cochrane Library from inception to November 2022. A snowball search on Google, Google Scholar, Research Gate, and MedRxiv, as well as bibliographic research, was performed to identify any other relevant articles. Observational studies that assessed the clinical usefulness of the RT-PCR technique in different human samples for the detection or screening of COVID-19 among patients or patient samples were considered for this review. The primary outcomes considered were sensitivity and specificity, while parameters such as positive predictive value (PPV), negative predictive value (NPV), and kappa coefficient were considered secondary outcomes. Results: A total of 85 studies out of 10,213 non-duplicate records were included for the systematic review, of which 69 articles were considered for the meta-analysis. The meta-analysis indicated better pooled sensitivity with the nasopharyngeal swab (NPS) than saliva (91.06% vs. 76.70%) and was comparable with the combined NPS/oropharyngeal swab (OPS; 92%). Nevertheless, specificity was observed to be better with saliva (98.27%) than the combined NPS/OPS (98.08%) and NPS (95.57%). The other parameters were comparable among different samples. The respiratory samples and throat samples showed a promising result relative to other specimens. The sensitivity and specificity of samples such as nasopharyngeal swabs, saliva, combined nasopharyngeal/oropharyngeal, respiratory, sputum, broncho aspirate, throat swab, gargle, serum, and the mixed sample were found to be 91.06%, 76.70%, 92.00%, 99.44%, 86%, 96%, 94.4%, 95.3%, 73.63%, and above 98; and 95.57%, 98.27%, 98.08%, 100%, 37%, 100%, 100%, 97.6%, and above 97, respectively. Conclusions: NPS was observed to have relatively better sensitivity, but not specificity when compared with other clinical specimens. Head-to-head comparisons between the different samples and the time of sample collection are warranted to strengthen this evidence.

## 1. Introduction

The recent global pandemic was caused by a respiratory tract infection in the Wuhan province of China in December 2019. The causative organism was recognized as a severe acute respiratory syndrome coronavirus 2 (SARS-CoV-2). The coronavirus disease 2019 (COVID-19) has spread across the world and contributed to many deaths in a huge proportion of the population. Fast and accurate detection of viruses and/or diseases is essential to controlling the sources of infection and having a better patient outcome through inhibiting disease progression [1,2].

According to the Foundation of Innovative New Diagnostics in collaboration with the WHO, the sensitivity and specificity of several available kits for molecular detection of SARS-CoV-2 by the PCR technique are around 92–100% and 98–100%, respectively [3]. However, the diagnostic accuracy of RT-PCR was studied in several reports, and it is less than the standard optimum value (100%) for an ideal diagnostic biomarker [4]. The RT-PCR showed false negative results in 3% of 167 confirmed cases of COVID-19 by chest CT typical criteria, which turned positive after repeating RT-PCR testing at an interval of about 5.0 ± 2.7 days [5]. Likewise, on a larger cohort of 1014 suspected COVID-19 patients, 88% of all studied cases showed positive chest CT findings of COVID-19, while only 59% had positive RT-PCR testing. Remarkably, 93% of that cohort turned into positive RT-PCR results within 5.1 ± 1.5 days after preliminarily being negatively tested, although they showed suggestive chest CT findings of COVID-19 [6]. In fact, RT-PCR detection is dependent on viral load, so it may show initial negative results during the incubation period, especially when the viral load is low [7].

According to experts, the results of real-time RT-PCR tests must be cautiously interpreted, along with the suggestive clinical presentations. Repeated tests can be considered when the clinical presentations resemble the diagnostic criteria of COVID-19 and the test is negative. A combination of objective evidence such as chest CT, C-reactive protein, and d-dimer, along with RT-PCR, could help in better patient management and outcomes [2].

The literature indicates that saliva is superior to the nasopharyngeal swab (NPS) for the detection of SARS-CoV-2, whereas other research evidence suggests that NPS may be more suitable than the oropharyngeal swab (OPS) for the detection of COVID-19 through RT-PCR. Hence, identifying the most suitable sample for the detection of disease, especially in the case of a pandemic, is crucial [8,9,10].

Although RT-PCR is considered the gold standard for COVID-19 diagnosis and is mandatory in our daily lives, there is variable evidence on the clinical performance when screening among various samples [11,12].

We, therefore, aimed to identify all the currently available literature and assess the clinical usefulness of RT-PCR in different COVID-19 samples through a comprehensive systematic literature review process and meta-analysis.

## 2. Materials and Methods

We followed pre-defined inclusion and exclusion criteria for selecting the studies in this review and adapted the Preferred Reporting Items for Systematic Reviews and Meta-Analyses (PRISMA) Guidelines to report this systematic review [13]. The protocol for this meta-analysis is submitted to The International Prospective Register of Systematic Reviews (PROSPERO) with a registration ID of CRD42023449573.

### 2.1. Criteria for Considering the Studies for This Review

The observational studies assessed the clinical usefulness of the RT-PCR technique in various human samples for the detection or screening of COVID-19 among the patients or patient samples that were considered for this review. Only the studies with full-text availability in the English language were considered. Studies comparing the numerous samples were also considered for this review. The primary outcomes considered were sensitivity and specificity, while parameters such as positive predictive value (PPV), negative predictive value (NPV), and kappa coefficient were considered secondary outcomes. We have considered all types of RT-PCR techniques in our review, per the author’s discretion. Any studies that used RT-PCR as a reference to assess the performance of other screening techniques were excluded. Studies such as reviews, descriptive studies, non-clinical studies, non-COVID-19 participants, commentary, guidelines, and qualitative analyses were excluded.

### 2.2. Search Methods for Identification of Studies

PubMed/Medline, Scopus, Embase, and Cochrane Library) were accessed through a comprehensive search strategy using all the possible keywords and entry terms from inception to November 2022. We also performed a snowball search on Google, Google Scholar Research Gate, and MedRxiv to identify any relevant articles. The reference lists of potential articles were also screened to identify additional potentially relevant citations. A detailed search strategy in various databases is provided in Appendix A.

### 2.3. Study Selection

All the identified records through a database literature search were retrieved in an Excel sheet and screened against the pre-defined criteria. The studies were screened by reading the title and abstracts in the initial stage, followed by the full text. Only the studies passing these two stages were considered for final inclusion in the review. Two independent reviewers were involved in the study selection to limit bias, and discrepancies were resolved through consensus or discussion with another member of the research team.

### 2.4. Data Extraction

The data were extracted to a well-defined data extraction form by two independent reviewers. The author’s first name and year of publication were used to identify the studies. The study detailed information such as year, country, study design, and study settings; the participants’ information including the total number of samples/participants, age and gender of cohort, and clinical presentation or characteristics; type of specimen; and the characteristics of RT-PCR techniques were captured from the studies. The outcomes were collected from the studies or calculated from the available data in terms of percentage with a 95% confidence interval. The highest values of primary and secondary outcomes were captured in the case of multiple RT-PCR kits used in the same study. Two independent reviewers were involved in the data extraction, and disagreements were resolved through discussion or consultation with another reviewer.

### 2.5. Evidence Synthesis and Meta-Analysis

All the evidence extracted through the systematic process was summarized narratively and presented in tabular form. The studies that have sufficient homogenous data or if there is a sufficient number of studies to perform meta-analysis were only considered for meta-analysis. Review Manager 5.4 was used to conduct the meta-analysis [14]. The available data were converted into percentage and standard error and presented as pooled outcomes with a 95% confidence interval. We used the random effect model, as there was substantial heterogeneity (I2 > 50%; *p* < 0.10) in all analyses. 

### 2.6. Publication Bias and Sensitivity Analysis

The visual inspection of the funnel plot for the sensitivity of RT-PCR in COVID-19 diagnosis was used to check publication bias using RevMan 5.4, which was further assessed for statistical significance with Egger’s and Begg’s test using comprehensive meta-analysis (trial version). A probability of less than 0.05 was considered to be statistically significant [15,16]. The sensitivity analysis was performed to check the robustness of the findings by removing the study with the lowest weight in the analysis [17].

## 3. Results

### 3.1. Study Selection Process

A total of 32,006 records were identified from literature sources and 10,213 records were screened by title and abstract following duplicate removal. A total of 8265 records (animal studies and case reports: 403; non-diagnostic and treatments: 303; guidelines and protocols: 42; non-English: 72; not RT-PCR: 6925; pediatric: 315; qualitative research and reviews: 205) were excluded at this stage, and the remaining 1948 full texts were considered for their eligibility. Following the exclusion of 1863 articles with numerous reasons (animal studies and case reports: 78; duplicate: 1; not outcome of interest: 1248; non-English: 43; not RT-PCR: 398; review: 95), 85 studies [18,19,20,21,22,23,24,25,26,27,28,29,30,31,32,33,34,35,36,37,38,39,40,41,42,43,44,45,46,47,48,49,50,51,52,53,54,55,56,57,58,59,60,61,62,63,64,65,66,67,68,69,70,71,72,73,74,75,76,77,78,79,80,81,82,83,84,85,86,87,88,89,90,91,92,93,94,95,96,97,98,99,100,101,102] were considered for this systematic review. Hence, a total of 69 articles with homogenous data were used for the meta-analysis. A detailed description of the study selection process is depicted in Figure 1.

### 3.2. Study Characteristics

The studies were published between the years 2020 and 2022 from different parts of the world with a major contribution from the USA, the UK, and India. The studies were observational making them retrospective, prospective, and cross-sectional in nature. The studies were from hospital settings or sample collection centers. The human samples were analyzed across the included studies. The majority of the studies included adult participants with an average age of 18 to 65 years. The participants were asymptomatic or symptomatic, severe or non-severe, and positive or negative at the time of sample collection. A detailed description of the studies and participant characteristics are provided in Table 1.

#### 3.2.1. Characteristics of RT-PCR Techniques

Many human samples such as the nasopharyngeal swab, oropharyngeal swab, respiratory tract specimens (bronchoalveolar lavage and broncho aspirates), throat, nasal, saliva, sputum, fecal, gargle, or mixed were used for the detection of COVID-19 using various RT-PCR techniques. The samples were stored at a cool temperature ranging from 40 °C to −800 °C. Many in-house and modified RT-PCR techniques were used by the studies by numerous companies. Detailed information on the RT-PCR techniques is depicted in Appendix A.

#### 3.2.2. The Diagnostic Parameters of RT-PCR in Various Samples

##### Nasopharyngeal Swabs

Sensitivity and specificity

A meta-analysis of 43 studies indicated a pooled sensitivity of 91.06% (95%CI: 88.91 to 93.21; I2: 100%) for nasopharyngeal swabs using RT-PCR techniques. (Figure 2A). A meta-analysis of 37 studies indicated a pooled specificity of 95.57% (95%CI: 95.19 to 95.96; I2: 100%) for nasopharyngeal swabs using different RT-PCR techniques. (Figure 2B).

PPV and NPV

A meta-analysis of 15 studies indicated a pooled PPV of 95.88% (95%CI: 91.59 to 100.16; I2: 100%) for nasopharyngeal swabs using numerous RT-PCR techniques. (Figure 2C). A meta-analysis of 15 studies indicated a pooled NPV of 91.58% (95%CI: 88.03 to 95.13; I2: 100%) for nasopharyngeal swabs using RT-PCR techniques. (Figure 2C).

Kappa coefficient

A meta-analysis of 16 studies indicated a pooled kappa coefficient of 0.79 (95%CI: 0.71 to 0.87; I2: 98%) for nasopharyngeal swabs using RT-PCR techniques. (Figure 2D). The diagnostic parameters of RT-PCR in the nasopharyngeal samples are provided in Figure 2.

### 3.3. Saliva Samples

#### 3.3.1. Sensitivity and Specificity

A meta-analysis of 14 studies indicated a pooled sensitivity of 76.70% (95%CI: 60.50 to 92.91; I2: 99%) in the saliva samples using RT-PCR techniques (Figure 3A). A meta-analysis of 11 studies indicated a pooled specificity of 98.27% (95%CI: 97.31 to 99.24; I2: 62%) in the saliva samples using various RT-PCR techniques (Figure 3B). 

#### 3.3.2. PPV and NPV

A meta-analysis of five studies indicated a pooled PPV of 90.16% (95%CI: 82.90 to 97.43; I2: 97%) in the saliva samples using RT-PCR techniques (Figure 3C). A meta-analysis of five studies indicated a pooled NPV of 90.37% (95%CI: 84.18 to 96.56; I2: 96%) in the saliva samples using RT-PCR techniques (Figure 3C). 

#### 3.3.3. Kappa Coefficient

A meta-analysis of 5 studies indicated a pooled kappa coefficient of 0.61 (95%CI: 0.44 to 0.79; I2: 98%) in the saliva samples using different RT-PCR techniques. (Figure 3D). The diagnostic parameters of RT-PCR in the saliva samples are provided in Figure 3.

### 3.4. Combined Nasopharyngeal/Oropharyngeal Samples

#### 3.4.1. Sensitivity and Specificity

A meta-analysis of 16 studies indicated a pooled sensitivity of 92.00% (95%CI: 87.57 to 96.43; I2: 100%) in the combined nasopharyngeal/oropharyngeal samples using different RT-PCR techniques (Figure 4A). A meta-analysis of 12 studies indicated a pooled specificity of 98.08% (95%CI: 96.64 to 99.52; I2: 100%) in the combined nasopharyngeal/oropharyngeal samples using RT-PCR techniques (Figure 4B). 

#### 3.4.2. PPV and NPV

A meta-analysis of five studies indicated a pooled PPV of 84.63% (95%CI: 70.14 to 99.12; I2: 100%) in the combined nasopharyngeal/oropharyngeal samples using RT-PCR techniques (Figure 4C). A meta-analysis of five studies indicated a pooled NPV of 96.12% (95%CI: 92.83 to 99.42; I2: 100%) in the combined nasopharyngeal/oropharyngeal samples using RT-PCR techniques (Figure 4C).

#### 3.4.3. Kappa Coefficient

A meta-analysis of five studies indicated a pooled kappa coefficient of 0.82 (95%CI: 0.67 to 0.98; I2: 98%) in the combined nasopharyngeal/oropharyngeal samples using various RT-PCR techniques (Figure 4D). The diagnostic parameters of RT-PCR in the nasopharyngeal and oropharyngeal samples are provided in Figure 4. The meta-analysis findings on various samples are provided in Table 2.

### 3.5. Respiratory Samples

Only two studies [26,69] reported the sensitivity and specificity of RT-PCR in respiratory samples. Wu et al., [26] indicated a sensitivity and specificity of 100% and Nakura Y et al. [69] reported a sensitivity and specificity of 99.44% and 100%, respectively. The studies by Wu S et al. [26] and Pekoz A et al. [41] recorded a PPV of 100% and 73.7; and an NPV of 100% and 100%, respectively. Additionally, Price T K et al. [83] recorded an NPV of 98% among their samples. Wu et al. [26] recorded a kappa coefficient of 1 among 52 samples analyzed. The details are provided in Table 3.

### 3.6. Sputum Samples

The study by Torres A et al. [33] indicated a sensitivity, specificity, PPV, NPV, and kappa coefficient of 86%, 37%, 38%, 85%, and 0.73, respectively. The study by Villota S D et al. [22] reported a sensitivity and specificity of 86% and 37%, respectively. The details are provided in Table 3.

### 3.7. Broncho Aspirate Samples

Only a single study by Pace V D et al. [23] used broncho aspirate samples for the detection of COVID-19 using RT-PCR. The sensitivity, specificity, and kappa coefficient were 96%, 100%, and 0.94, respectively. The details are provided in Table 3.

### 3.8. Throat Swab Samples

Two studies [30,95] used throat samples for the detection of COVID-19 by RT-PCR. The study by Wang B et al. [30] reported a sensitivity, specificity, PPV, NPV, and kappa coefficient of 97.62%, 100%, 100%, 98.52%, and 0.985, respectively. The study by Lu Y et al. [95] reported a sensitivity, specificity, PPV, NPV, and kappa coefficient of 94.4%, 100%, 100%, 99%, and 0.996, respectively. The details are provided in Table 3.

### 3.9. Gargle Samples

The study by Dumaresq J et al. [43] reported a sensitivity and kappa coefficient of 95.3% and 0.94, respectively, in gargle samples for the detection of COVID-19 using the RT-PCR technique. The details are provided in Table 3.

### 3.10. Serum Samples

Ramírez AM et al. [96] reported a sensitivity, specificity, PPV, NPV, and kappa coefficient of 73.63%, 97.6%, 96.73%, 75%, and 0.69, respectively. The details are provided in Table 3.

### 3.11. Mixed Samples

The sensitivity and specificity were reported by four studies using mixed samples for the detection of COVID-19. The studies by Ferreira BLS et al. [42], Omar S et al. [85], Yip CCY et al. [87], and Yang M et al. [97] reported a sensitivity of 98.04%, 95%, 99.1%, and 100%, respectively. Similarly, the specificity was observed to be 100%, 97%, 100%, and 100%, respectively. The PPV and NPV were reported by three studies [42,85,97], and they were 100% and 93.3%, 82.4, and 99.9%, and 100% for studies by Ferreira BLS et al. [42], Omar S et al. [85], and Yang M et al. [97], respectively. The kappa coefficient was reported by only one study, Ferreira BLS et al. [42], and it was 0.96. The details are provided in Table 3.

### 3.12. Publication Bias

A visual inspection of the funnel plot reveals an obvious asymmetry, which represents the chances of publication bias. This was confirmed statistically by Egger’s test (*p* = 0.00003) but not Begg’s test (*p* = 0.0982). The funnel plot is provided in Appendix A.

### 3.13. Sensitivity Analysis

The sensitivity analysis was performed by altering the analysis model from the random effect model to the fixed effect model on NPS sensitivity analysis (Figure 2A). This made a small change in the overall effect measure, which is 91.06% (95%CI: 88.91 to 93.21) in the random effect model and 94.53% (95%CI: 94.53 to 94.54) in the fixed effect model. The sensitivity analysis result is provided in Appendix A.

## 4. Discussion

COVID-19 can manifest in a variety of forms ranging from simple flu-like illness to death [103]. Various samples are used for the diagnosis of COVID-19 using many techniques, including RT-PCR. The diagnostic performance of various sampling approaches needs to be investigated to gain a better picture of all these aspects [104].

Our review provided evidence that pharyngeal samples (combined nasopharyngeal/oropharyngeal) have an equivalent sensitivity to nasopharyngeal samples, whereas saliva samples have a lesser sensitivity compared to the two other types of samples. A previous systematic review reported a comparable diagnostic performance with pooled nasal and throat swabs in comparison with nasopharyngeal swabs, which is considered to be the gold standard technique. Moreover, the self-collection of samples has influenced diagnostic accuracy [104].

As indicated in our review, respiratory samples, combined nasopharyngeal/oropharyngeal samples, broncho aspirate samples, throat swab samples, gargle samples, and mixed samples had better sensitivity than other samples, like serum and saliva, compared to nasopharyngeal swabs. Similarly, the study by Becker et al. recorded that saliva had approximately 30% lesser sensitivity than NPS, and it was 50% less sensitive in those cohorts of samples taken less than 21 days from the first symptom occurrence [105]. Similar findings were observed in the previous meta-analysis by Lee et al. [106].

Combined NPS/OPS, saliva, respiratory, broncho aspirate, throat, and mixed samples had better specificity than NPS. The current review indicates lower specificity with NPS than other specimens except for the sputum sample, which had reduced specificity compared to NPS. A community study by Torres et al. reported that saliva had 99.1% relative specificity to NPS [107]. Better diagnostic accuracy and specificity with saliva samples have been reported by many other studies [108,109,110]. Moreover, another study by Sasikala et al. reported that there was no difference between the diagnostic performance of saliva samples collected by healthcare workers and the patients themselves [111].

This review suggests that throat and respiratory samples had a similar positive predictive value (PPV) compared to NPS, while all other specimens had a lower PPV than NPS. Wang H et al. also found that NPS had better performance than other samples and recommended it as the best specimen for detecting COVID-19 through RT-PCR techniques [112]. The findings from this study can be used to develop protocols and guidelines for diagnosing COVID-19 and similar infections. Although NPS is considered the gold standard for diagnosing COVID-19, other samples have also been found to be equally helpful. Head-to-head analysis between different sampling strategies and specimens needs to be studied to develop the best alternative, cost-effective, and accurate diagnostic techniques.

This review had some limitations. First, there was a significant level of heterogeneity in all the meta-analyses performed, so caution should be taken when interpreting the findings. Second, English language restriction might have contributed to the exclusion of studies. However, comprehensive literature searches in all the available databases helped to collate the maximum possible information. Third, the variation in the RT-PCR techniques used and their processes might have contributed to the findings. Fourth, there was a lack of information with respect to sampling techniques and time of sampling. Hence, further research studies should focus on this. Future meta-analyses that emphasize subgroup analysis based on COVID-19 status, severity, and other important parameters should be planned.

## 5. Conclusions

The current meta-analysis suggests that NPS has a better or similar sensitivity than other samples, especially the specimens collected from any parts of the respiratory system, while the relative specificity of NPS was lower compared to other samples. Caution should be taken while interpreting the results due to the high heterogeneity in the analyses.

## Figures and Tables

**Figure 1 diagnostics-13-03057-f001:**
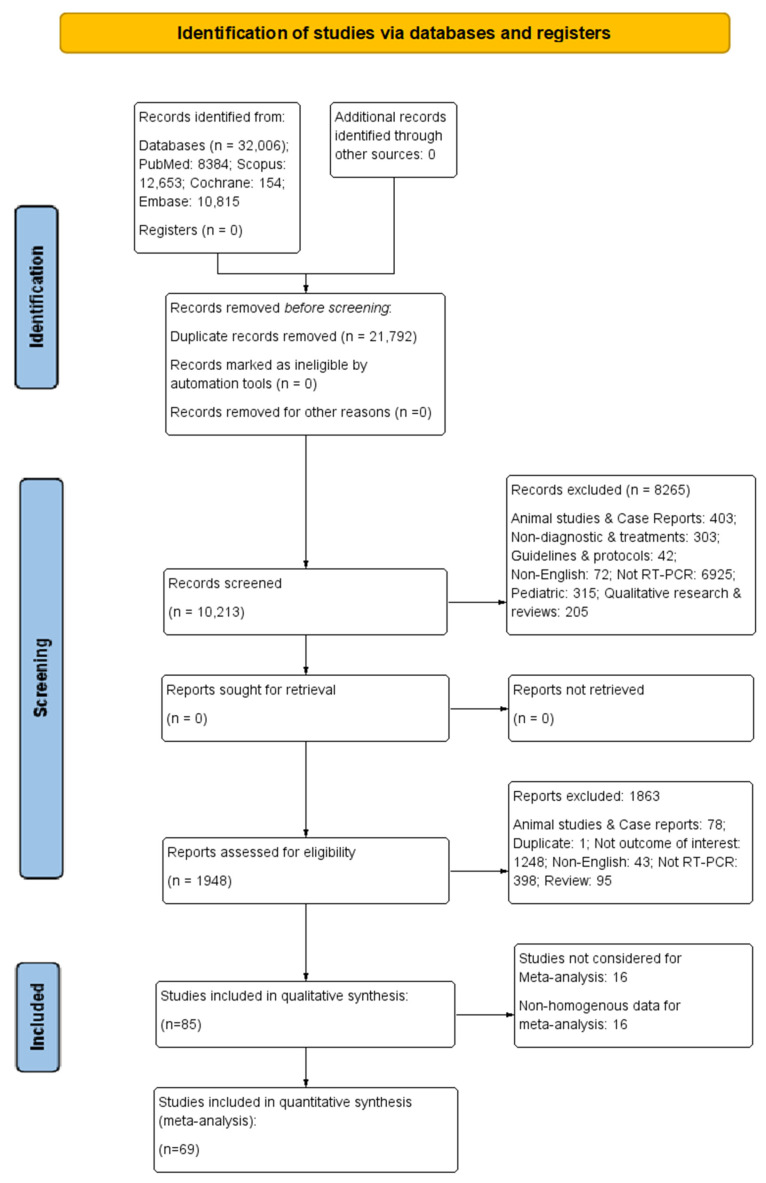
The PRISMA flow diagram for study selection.

**Figure 2 diagnostics-13-03057-f002:**
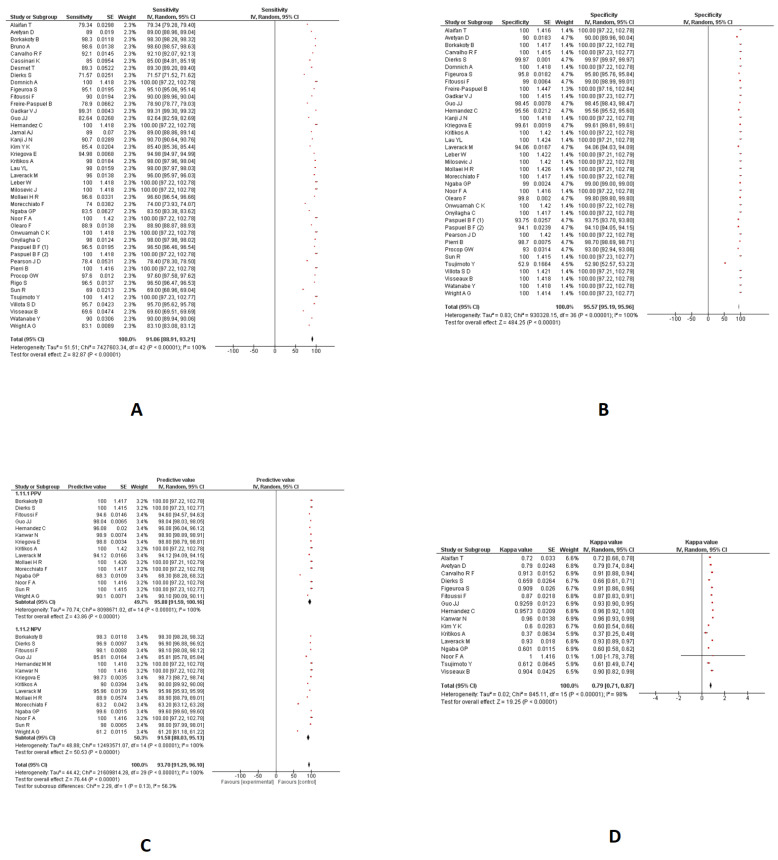
The diagnostic parameters of RT-PCR in the nasopharyngeal samples. (**A**): Pooled Sensitivity; (**B**): Pooled Specificity; (**C**): Pooled PPV and NPV; (**D**): Pooled kappa coefficient.

**Figure 3 diagnostics-13-03057-f003:**
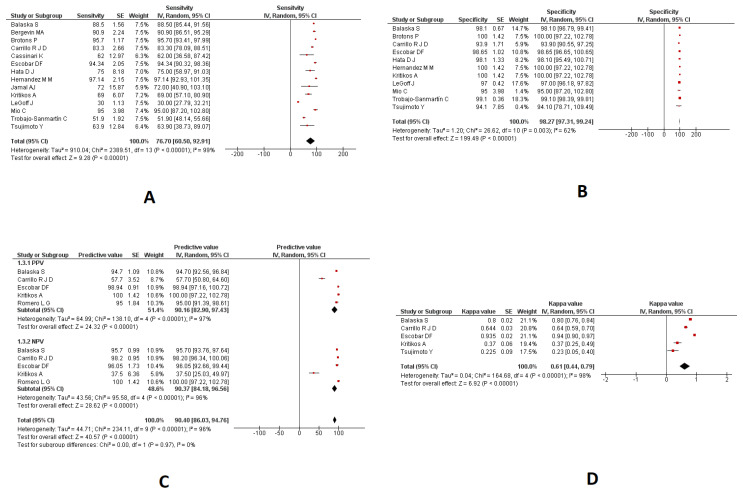
The diagnostic parameters of RT-PCR in saliva samples. (**A**): Pooled Sensitivity; (**B**): Pooled Specificity; (**C**): Pooled PPV and NPV; (**D**): Pooled kappa coefficient.

**Figure 4 diagnostics-13-03057-f004:**
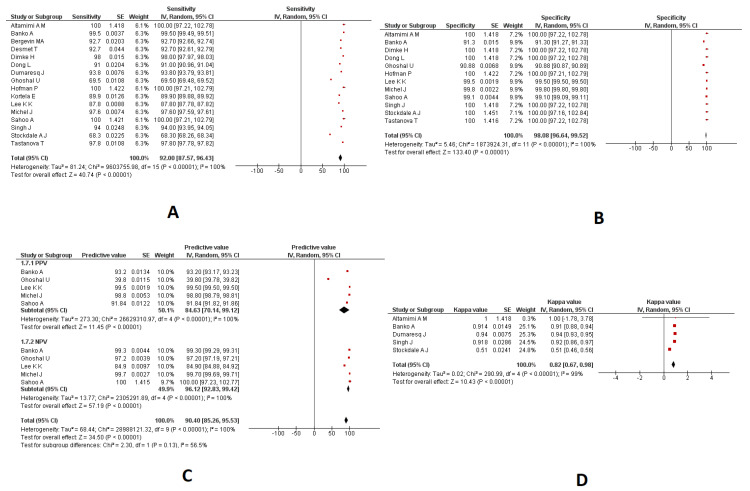
The diagnostic parameters of RT-PCR in the combined nasopharyngeal/oropharyngeal samples. (**A**): Pooled Sensitivity; (**B**): Pooled Specificity; (**C**): Pooled PPV and NPV; (**D**): Pooled kappa coefficient.

**Table 1 diagnostics-13-03057-t001:** Characteristics of the included studies and patients.

Study ID, Year, Country	Study Design	Study Settings	Total Number of Samples/ Participants	Age of Cohort	Male/ Female	Clinical Presentation/ Characteristics
Escobar et al., 2021; Chile [18]	Cohort selection of a cross-sectional study	Multi-specialized Guillermo Grant Benavente Hospital (HGGB) and three Family Health Centers (FHCs) in the Chilean city of Concepción	127 saliva and 127 NPS	<18: 5; 18–34: 69; 35–50: 29; 51–65: 20; >65: 4	68/59	Symptomatic: 111; Asymptomatic:15; No information: 1; RT-PCR positive: 104; RT-PCR negative: 150
Singh et al., 2021; India [19]	Quality audit study	Medical college institution	92 samples and 60 controls	NR	NR	RT-PCR positive: 92; Healthy individuals: 30; Other respiratory disease: 30
Figueroa et al., 2021; Ecuador [20]	Case–control study	NR	242 clinical specimens and 11 negative controls	NR	NR	122 SARS-CoV-2-positive and 120 SARS-CoV-2-negative
LeGof et al., 2021; France [21]	Prospective observational study	Two community COVID-19 screening centers	1718	37 (26–52) a	774/944	+NPS RT-PCR: 117; Symptomatic: 530
Villota et al., 2021; Ecuador [22]	Cross-sectional study	Two centers from Ecuador and USA	192 clinical samples (NPS: 132; sputum: 60)	NR	NR	Positive: 142; Negative: 50
De Pace et al., 2021; Italy [23]	Consecutive prospective observational study	Intensive Care Units of San Martino Hospital (Genoa, Italy	75 patients	65 (31–81) b	56/19	BAS: 43 (57.3%); Negative: 30.2%; Positive: 69.8%; BAL: 32 (42.7%); Negative: 37.5%; Positive: 62.5%
Kanwar et al., 2021; USA [24]	Prospective salvage sample study	University of Kansas Health System (TUKHS)	201 samples	57 (15–92) a	103/98	Positive: 99; Negative: 102
Michel et al., 2021; Germany [25]	NR	Robert Koch Institute	424 specimens	NR	NR	Positive: 424
Wu et al., 2021; China [26]	Cross-sectional study	Shenzhen Third People’s Hospital and a compulsory quarantine facility	52 (throat: 30; nasal: 7; NPS: 7; sputum: 8	NR	NR	Positive: 26; Negative: 26
Lee et al., 2021; UK [27]	Prospective, multi-center, cohort study	Secondary and tertiary care hospitals in Scotland	1368 patients with 3822 tests	68 (53–80) b	731/637	Confirmed positive: 496
Borkakoty et al., 2021; India [28]	NR	State of Assam	240 random samples	NR	NR	Positive: 120; Negative: 120
Hata et al., 2021; USA [29]	NR	Mayo Clinic	135 participants	20–83 c	NR	Positive: 28; Negative: 106
Wang et al., 2020; China [30]	NR	The Second Xiangya Hospital	242 samples	NR	NR	Positive: 42 (34 throat swabs and 8 fecal samples); Negative: 200
Mollaei et al., 2020; Iran [31]	NR	Kerman Reference Laboratory	30 infected patients	NR	NR	Varies based on the gene chosen
Pierri et al., 2022; Italy [32]	Post-analysis of a GENCOVID study	GENCOVID people in direct contact with positive patients from the Campania region, Italy	258 samples	NR	NR	Positive: 164
Torres et al., 2021; Ecuador [33]	Descriptive-correlating, retrospective, cross-sectional study	Santo Domingo General Hospital (Santo Domingo de los Tsáchilas, Ecuador)	773 samples	1–14 years: 38; 15–19 years: 33; 20–49 years: 461; 50–64 years: 127; >65 years: 74	344/389	Symptomatic: 515; Asymptomatic: 218
Pearson et al., 2021; Canada [34]	NR	MSH/UHN clinical diagnostics lab	59 samples	NR	NR	Positive: 29; Negative: 30
Kriegova et al., 2021; Czech Republic [35]	Large prospective cohort	University Hospital Olomouc and Sumperk Hospital, Czechia	1038 subjects	NR	NR	Positive: 297; Negative 741
Onyilagha et al., 2021; Canada [36]	Cross-sectional study	NR	90 samples	NR	NR	Negative: 40; Positive: 50
Desmet et al., 2021; Belgium [37]	Prospective observational study	Ghent University Hospital	36 patients	61 (22–90) b	21/25	NP or OP/N positive: 35; Combined positive: 31; Mild: 7; Moderate: 10; Severe: 13; Critical care: 5; Pre-symptomatic: 1
Kanji et al., 2021; Canada [38]	Prospective cross-sectional study	Province of Alberta, Canada	49 patients	72 (25–97) b	15/34	Positive: 49; Negative: 52
Gómez-Romero et al., 2021; Mexico [39]	Prospective database study	Epidemiology department of the Health Ministry of the State of Morelos (Secretaría de Salud Morelos, SSM)	140 healthcare workers/sample	NR	NR	Positive: 36; Negative: 104
Milosevic et al., 2021; United States [40]	Prospective cohort study	Penn State Health Milton S. Hershey Medical Center	60 samples	NR	NR	Positive: 30; Negative: 30
Pekosz et al., 2021; United States [41]	Prospective cohort study	FDA EUA study samples which occurred across 21 geographically diverse study sites	251 sample			Symptomatic: 251
Ferreira et al., 2021; Brazil [42]	Prospective cohort study	State of Rio de Janeiro and the state of Ceará	65 patients	NR	NR	NPS: 42; Serum: 12; Saliva: 11; Positive: 51; Negative: 14
Dumaresq et al., 2021; Canada [43]	Prospective cohort study (SPRING study)	Département de microbiologie et d’infectiologie du centre hospitalier universitaire	2010 sample from 987 patients	40 (6–91) a	NR	1005 ONPS and 1005 gargles; Symptomatic: 987; Asymptomatic: 987
Morecchiato et al., 2021; Italy [44]	Prospective cohort study (SPRING study)	Microbiology and Virology Unit of Florence Careggi University Hospital (Florence, Italy)	139 samples	NR	NR	Positive: 96; Negative: 43
Olearo et al., 2021; Germany [45]	Cross-sectional retrospective study	University Hospital Hamburg.	7513 HCWs (55,122 samples); 11,192 sample pools	NR	NR	Negative: 11,041; Invalid: 82; Positive: 69
Ghoshal et al., 2021; India [46]	Retrospective observational study	Triage of a dedicated COVID-19 tertiary care center with 180 beds including 30 ICU ventilator beds	1807 patients	NR	NR	RT-PCR positive; 174; TrueNat: 174
Balaska et al., 2021; Greece [47]	Prospective observational study	AHEPA University Hospital, Thessaloniki	420 pairs of samples	44.7 (13) a	161/259	Positive diagnostic sample: 27.7%; Screening sample: 5%
Watanabe et al., 2021; Japan [48]	NR	Kawasaki Rinko General Hospital and the Matsudo City General Hospital	96 patients	49.3 (27.8) a	45/51	Positive: 20; Negative: 76
Domnich et al., 2021; Italy [49]	Prospective observational study	San Martino Policlinico Hospital (Genoa, Liguria, Northwest Italy)	98 samples	NR	NR	Positive: 98
Kim et al., 2021; South Korea [50]	NR	Kyungpook National University	300 samples	NR	NR	Positive: 260; Negative: 40
Carvalho et al., 2021; Brazil [51]	NR	Municipal medical service	346 samples	NR	NR	Detectable: 194; Undetectable: 152
Kritikos et al., 2021; Switzerland [52]	Prospective observational study	Tertiary university hospital in Lausanne, Switzerland	58 patients	70 (61–77) b	45/13	Symptomatic: 49
Brotons et al., 2021; Spain [53]	Three-phase cross-sectional study	Molecular Microbiology Department of Sant Joan de Déu Hospital	183 samples	NR	NR	Positive: 10; Negative: 173
Laverack et al., 2021; USA [54]	NR	Cornell COVID-19 Testing Laboratory by three other COVID-19 testing laboratories in the United States	225 samples	NR	NR	NPS: 201; AN: 24; NPS positive: 100; Negative: 101; AN positive: 12; AN negative: 12
Avetyan et al., 2021; Armenia [55]	Cross-sectional study	Institute of Molecular Biology, National Academy of Sciences	NPS: 74; RNA sample: 196	NR	NR	NPS: Positive: 44; Negative: 30; RNA sample positive: 196
Hernandez et al., USA; 2021 [56]	NR	Clinical Microbiology Laboratory at the Mount Sinai Health System	60 patients	NR	NR	NR
Hernández et al., 2021; Colombia [57]	NR	Not reported	94 samples			Positive: 49; Negative: 45
Leber et al., 2021; UK [58]	Prospective cohort study	GP participating in the National Influenza Surveillance Network in the ski resort of Schladming-Dachstein	66 patients	NR	NR	Positive: 22; Negative: 44
Gadkar et al., 2021; Canada [59]	NR	Microbiology and virology laboratories of BC Children’s Hospital	372 samples	NR	NR	Positive: 142
Bruno et al., 2021; Ecuador [60]	NR	INSPI and UDLA	1036 samples	NR	NR	Positive: 543; Negative: 493
Sun et al., 2021; France [61]	Single center, retrospective, observational study	Radiation therapy department, Gustave Roussy, Paris-Saclay University	480 patients	62 (50–70) b	228/252	Positive: 26; Negative: 446
Rigo et al., 2021; Pordenone [62]	NR	Microbiology and Virology Department Laboratory	180 samples	NR	NR	Positive: 93; Negative: 88
Banko et al., 2021; Serbia [63]	NR	Laboratory of Molecular Microbiology, Institute for Biocides and Medical Ecology, Belgrade	354 samples	NR	NR	Sansure Biotech: Positive: 190; Negative: 164 GeneFinderTM: Positive: 176; Negative: 178 TaqPathTM: Positive: 178; Negative: 176
Tastanova et al., 2021; Switzerland [64]	NR	University Hospital Zurich and at ADMed Laboratory in La Chaux-de-Fonds, Switzerland	184 samples	NR	NR	Positive: 92; Negative: 92
Noor et al., 2021; Bangladesh [65]	Case–control sample study	Department of Biochemistry and Molecular Biology	240 samples	NR	NR	Positive: 120; Negative: 120
Fitoussi et al., 2021; France [66]	Prospective observational study	entre Cardiologique du Nord-CCN, Saint-Denis, France	239 patients	NR	NR	Positive: 140; Negative: 99
Freire-Paspuel et al., 2020; Ecuador [67]	Prospective observational study	Laboratory of “Universidad de Las Américas” in Quito (Ecuador)	89 samples	NR	NR	Positive: 57; Negative: 32
Dierks et al., 2021; Germany [68]	NR	University Medical Center Göttingen	322 samples	NR	NR	Positive: 21; Negative: 301
Nakura et al., 2021; Japan [69]	NR	Osaka Women’s and Children’s Hospital, Osaka Habikino Medical Center, and Osaka General Medical Center of the Osaka Prefectural Hospital	213 samples	NR	NR	Sputum: 35; NPS: 124; Saliva: 7
Stockdale et al., 2021; UK [70]	NR	Liverpool University Hospitals NHS Foundation Trust	429 patients	67 (55–78) b	257/172	Positive: 293; Negative: 136
Kortela et al., 2021; Finland [71]	Population-based retrospective study	Helsinki Capital Region, Finland	3008 patients	52.5 (19.7) a; 51 (36–69) b	1215/1794	Not suspected: 514; Not excluded: 1318; High suspicion: 516; Laboratory confirmed: 574; Not known: 86; Positive: 585; Negative: 2246
Altamimi et al., 2021; Saudi Arabia [72]	NR	Saudi Center for Disease Prevention and Control (SCDC) Laboratories	94 samples	NR	NR	Positive: 63; Negative: 31
Visseaux et al., 2021; France [73]	NR	Virology Laboratory of Bichat-Claude Bernard University Hospital, Paris, France	94 samples	NR	NR	Positive: 69; Negative: 25
Cassinari et al., 2021; France [74]	Prospective observational study	Rouen University Hospital	130 patients	NR	NR	Positive: 13; Negative: 117
Carrillo et al., 2021; Manila [75]	Prospective cross-sectional diagnostic accuracy study	Philippine General Hospital	197 patients	32 (22–64)	74/123	Positive: 18; Negative: 179
Girish et al., 2021; India [76]	Cross-sectional, analytical study	BJ Medical College and Civil Hospital	309 patients	NR	NR	Positive: 55; Negative: 254
Freire-Paspuel et al., 2021; Ecuador [77]	NR	NR	97 samples	NR	NR	Positive: 43; Negative: 54
Dong et al., 2021; China [78]	NR	Hospitalized patients or close contacts of hospitalized patients tested by Beijing CDC (BJCDC), Wuhan CDC (WHCDC), and a government-designated clinical test laboratory	196 samples	NR	NR	Febrile suspected patients: 103; Close contacts: 77; Convalescents: 16; Positive: 132; Negative: 64
Gupta-Wright et al., 2021; UK [79]	Retrospective cohort study	Two hospitals within an acute NHS Trust in London, UK	4008 patients	69 (56–81) b*	1142/651 *	Non-COVID-19: 2215; COVID-19 diagnosis: 1793; Positive: 1391; Negative: 283
Dimke et al., 2021; Denmark [80]	NR	Department of Clinical Microbiology, Odense University Hospital	87 samples	NR	NR	Positive: 57; Negative: 30
Alaifan et al., 2021; Saudi Arabia [81]	NR	Diagnostic laboratories at the Saudi Center for Diseases Control and Prevention	185 samples	NR	NR	Positive: 121; Negative: 64
Onwuamah et al., 2021; Nigeria [82]	Retrospective study	The Nigerian Institute of Medical Research from people living in Lagos, Nigeria	63 samples	NR	NR	Positive: 48; Negative: 15
Price et al., 2021; USA [83]	Prospective observational study	University of California, Los Angeles Health System	10,165 samples from 8948 patients	NR	NR	NPS: 10,215; Bronchoalveolar lavage: 121; Expectorated sputum: 22; Miscellaneous sample types: 35; Positive: 630; Negative: 9535
Trobajo-Sanmartín et al., 2021; Spain [84]	Prospective study	Clinical microbiology department of the Navarra Hospital Complex	674 pairs of samples (NP and saliva)	36 (19) b	300/374	Positive: 337; Negative: 337; Symptomatic: 333; Non-symptomatic: 341
Omar et al., 2021; South Africa [85]	Retrospective descriptive cross-sectional study	Data from the mobile COVID-19 PCR testing laboratory database and the non-COVID-19 ICU database	315 samples from 1032 patients	40 (20.4) a	551/481	NPS: 281 Nasal swab: 17; OPS: 1; Tracheal respirate: 7; Not specified: 13; Positive: 51; Negative: 264
Bergevin et al., 2021; Canada [86]	Prospective evaluation	Laval region of Quebec, Canada	773 pairs	Positive: 44 (31–58) b	Positive: 80/85	Positive: 165 (symptomatic: 148; asymptomatic: 17)
Yip et al., 2021; China [87]	NR	The University of Hong Kong-Shenzhen Hospital	296 samples	NR	NR	Positive: 105; Negative: 191
Renzoni et al., 2021; Switzerland [88]	Retrospective analysis	Geneva University Hospitals	61 samples	NR	NR	Positive: 61; Control: 16
Tsujimoto et al., 2021; Japan [89]	Single-center, prospective study	National Centre for Global Health and Medicine (Tokyo, Japan)	10 patients (57 sets of NPS, NS, and SS samples)	47 (30–70) b	2/8	Positive: 48; Negative: 9
Mio et al., 2021; Italy [90]	NR	Department of Laboratory Medicine, University Hospital of Udine, Italy	30 patient samples	NR	NR	Positive: 19; Negative: 11
Lau et al., 2021; Malaysia [91]	NR	Hospital Sungai Buloh, Malaysia	113 samples	NR	NR	Positive: 78; Negative: 35
Shen et al., 2021; China [92]	NR	Beijing Center for Disease Prevention and Control (BJCDC)	142 samples	NR	NR	Kit I: Positive: 130; Negative: 12; Kit II: Positive: 116; Negative: 26; Kit III: Positive: 114; Negative: 28; Kit IV: Positive: 129; Negative: 13
Freire-Paspuel et al., (B) 2020; Ecuador [93]	NR	Laboratory of “Universidad de Las Américas” in Quito (Ecuador)	48 samples	NR	NR	Positive: 30; Negative: 18
Guo et al., 2020; China [94]	NR	Three centers in China	500 subjects	0.75–93c	258/242	Positive: 242; Negative: 258; OPS: 395; Sputum: 167
Lu et al., 2020; China [95]	NR	Liuzhou People’s Hospital	118 patients	Cases: 35.94 (16.32); Control: 36.50 (19.93) a	72/46	COVD-19: 18; Control: 100
Martín Ramírez et al., 2022; Spain [96]	Retrospective cohort study	Princesa University Hospital	303 patients	Pre-pandemic control: 73.5 (62.5–85.5); Pandemic control: 69 (62–83); Positive: 64 (56–72) b	Pre-pandemic control: 25/25; Pandemic control: 32/18; positive: 139/64	Positive: 203; Pre-pandemic control: 50; Pandemic control: 50;
Yang et al., 2022; China [97]	NR	Department of Laboratory Medicine, Shengjing Hospital of China Medical University,	63 samples	NR	NR	Positive: 28; Negative: 35
Sahoo et al., 2021; India [98]	Cross-sectional observational study	Department of Microbiology, ABVIMS, and Dr. RML Hospital	500	NR	NR	Positive: 49; Negative: 451
Hofman et al., 2021; France [99]	Prospective cohort study	Downtown free screening centers available to the population of the Nice metropolitan area and the outpatient clinic of the Department of Pulmonary Medicine of the University Hospital of Nice	112 samples/subjects	40 (15) b	69/43	Positive: 45; Negative: 67
Jamal et al., 2020; Canada [100]	Population-based surveillance of consecutive patients	Six Toronto Invasive Bacterial Disease Network	91 patients	66 (23–106) b	52/39	Positive: 72; Negative: 19
Ngaba et al., 2021; Cameroon [101]	Cross-sectional and comparative study	Douala Gynaeco-Obstetrics and Pediatric Hospital molecular biology laboratory	1810 patients	0–71 + c	1226/559	NPS: 1736; Saliva: 2; Throat swab: 1; Positive: 35; Negative: 1775
Procop et al., 2020; USA [102]	NR	Cleveland Clinic	239 samples	49.28 (16.86) a	NR	Positive: 168; Negative: 71

AN: anterior nares; HCW: healthcare worker; ICMR: Indian Council for Medical Research; INSPI: Instituto Nacional de Salud Pública e Investigación Leopoldo Izquieta Pérez; NIV: National Institute of Virology; NPS: nasopharyngeal swab; NR: not reported; UDLA: Universidad de Las Américas. a indicates mean; b indicates median; c indicates range. * indicates COVID-19-diagnosed patients.

**Table 2 diagnostics-13-03057-t002:** The meta-analysis findings on the diagnostic parameters of RT-PCR in various samples.

Parameter	Number of Studies	Pooled Effect Measure (95%CI)	Heterogeneity
** *Nasopharyngeal swabs* **			
Sensitivity	43	91.06% (95%CI: 88.91 to 93.21)	100%
Specificity	37	95.57% (95%CI: 95.19 to 95.96)	100%
PPV	15	95.88% (95%CI: 91.59 to 100.16)	100%
NPV	15	91.58% (95%CI: 88.03 to 95.13)	100%
Kappa coefficient	16	0.79 (95%CI: 0.71 to 0.87)	98%
** *Saliva samples* **			
Sensitivity	14	76.70% (95%CI: 60.50 to 92.91)	99%
Specificity	11	98.27% (95%CI: 97.31 to 99.24)	62%
PPV	5	90.16% (95%CI: 82.90 to 97.43)	97%
NPV	5	90.37% (95%CI: 84.18 to 96.56)	96%
Kappa coefficient	5	0.61 (95%CI: 0.44 to 0.79)	98%
** *Combined nasopharyngeal/oropharyngeal samples* **			
Sensitivity	16	92.00% (95%CI: 87.57 to 96.43)	100%
Specificity	12	98.08% (95%CI: 96.64 to 99.52)	100%
PPV	5	84.63% (95%CI: 70.14 to 99.12)	100%
NPV	5	96.12% (95%CI: 92.83 to 99.42)	100%
Kappa coefficient	5	0.82 (95%CI: 0.67 to 0.98)	98%

**Table 3 diagnostics-13-03057-t003:** The diagnostic parameters of RT-PCR in different samples.

Parameter	Study	Total Participants	Effect Measure
** *Respiratory samples* **			
Sensitivity	Wu S et al. [26]	52	100
	Nakura Y et al. [69]	213	99.44
Specificity	Wu S et al. [26]	52	100
	Nakura Y et al. [69]	213	100
PPV	Wu S et al. [26]	52	100
	Pekoz A et al. [41]	251	73.7
NPV	Wu S et al. [26]	52	100
	Pekoz A et al., [41]	251	100
	Price T K et al. [83]	10,165	98
Kappa Coefficient	Wu S et al. [26]	52	1
** *Sputum samples* **			
Sensitivity	Villota S D et al. [22]	50	90
	Torres A et al. [33]	229	86
Specificity	Villota S D et al. [22]	50	100
	Torres A et al. [33]	229	37
PPV	Torres A et al. [33]	229	38
NPV	Torres A et al. [33]	229	85
Kappa Coefficient	Torres A et al. [33]	229	0.73
** *Broncho aspirate samples* **			
Sensitivity	Pace V D et al. [23]	75	96
Specificity	Pace V D et al. [23]	75	100
Kappa Coefficient	Pace V D et al. [23]	75	0.94
** *Throat swab samples* **			
Sensitivity	Wang B et al. [30]	42	97.62
	Lu Y et al. [95]	18	94.4
Specificity	Wang B et al. [30]	158	100
	Lu Y et al. [95]	18	100
PPV	Wang B et al. [30]	200	100
	Lu Y et al. [95]	18	100
NPV	Wang B et al. [30]	200	98.52
	Lu Y et al. [95]	18	99
Kappa Coefficient	Wang B et al. [30]	200	0.985
	Lu Y et al. [95]	18	0.996
** *Gargle samples* **			
Sensitivity	Dumaresq J et al. [43]	1005	95.3
Kappa Coefficient	Dumaresq J et al. [43]	1005	0.94
** *Serum samples* **			
Sensitivity	Ramírez AM et al. [96]	265	73.63
Specificity	Ramírez AM et al. [96]	265	97.6
PPV	Ramírez AM et al. [96]	265	96.73
NPV	Ramírez AM et al. [96]	265	75
Kappa Coefficient	Ramírez AM et al. [96]	265	0.69
** *Mixed samples* **			
Sensitivity	Silva ferreira B I et al. [42]	65	98.04
	Omar S et al. [85]	319	95
	Yip CCY et al. [87]	106	99.1
	Yang M et al. [97]	35	100
Specificity	Silva ferreira B I et al. [42]	65	100
	Omar S et al. [85]	319	97
	Yip CCY et al. [87]	106	100
	Yang M et al. [97]	28	100
PPV	Silva ferreira B I et al. [42]	65	100
	Omar S et al. [85]	319	82.4
	Yang M et al. [97]	63	100
NPV	Silva ferreira B I et al. [42]	65	93.3
	Omar S et al. [85]	319	99.9
	Yang M et al. [97]	63	100
Kappa Coefficient	Silva ferreira B I et al. [42]	65	0.96

PPV: positive predictive value; NPV: negative predictive value.

## Data Availability

All the data related to this paper is provided in the text or as a Appendix A along with this paper. Any additional information can be made available from the corresponding author upon request.

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
