# Peer review of "The Diagnostic Performance of Various Clinical Specimens for the Detection of COVID-19: A Meta-Analysis of RT-PCR Studies"

_diagnostics, 2023, doi:10.3390/diagnostics13193057_

Round 1

Reviewer 1 Report

Pls see my comments in the attached manuscript - Reviewed

Needs moderate editing.

Also noticed redundant sentences that need to be minimized to avoid reader fatigue.

Author Response

Dear Editor/Reviewer,

Thank you so much for your useful comments and suggestions on our manuscript entitled “Diagnostic Performance of Various Clinical Specimens for the Detection of Covid-19: A Meta-Analysis of RT-PCR Studies”. We have modified the manuscript accordingly using tracked changes, and detailed corrections are listed (see attached file).

Reviewer 2 Report

This paper describes an ever current topic, important for this but future pandmics of similar type too. However, beside the language (grammar, style and repetition of words), there are some issues that should be corrected by the authors:

1. Recommendation for the Title:

Diagnostic Performance of Various Clinical Specimens for the Detection of Covid-19 by RT-PCR: A Meta-Analysis

Instead of

Diagnostic Performance of Various Clinical Specimens for the Detection of Covid-19: A Meta-Analysis of RT-PCR Studies

2. ABSTRACT

Row 38 – change capital letter C into small one in the sentence: “The Sensitivity and specificity of samples such as 27 nasopharyngeal swabs, saliva, Combined nasopharyngeal/oropharyngeal, respiratory, sputum, 28 broncho aspirate, throat swab…”

3. I would suggest the authors to avoid absolute definitions where it is not adequate: Line 38: "respiratory infection" , please change to "systemic infection with respiratory transmission" for example. 

4. MATERIALS AND METHODS

Rows 86-97: Please define inclusion criteria following PICOS system.

Rows 99-105: remove links from the text

It is strongly recommended to report complete last search strategy instead of step by step system. Please change the way of reporting of search strategy in Supplementary file S1.

Rows 114-117: Need to reword according to the successive steps in systematic review process like this

I - All the identified records through database literature search and other resources were retrieved to an Excel sheet

II - The duplicate files were removed.

III – Title and abstracts of remaining articles were screened

IV - Highly irrelevant articles as per the inclusion and exclusion criteria were excluded at this stage.

Rows 118-120: Reword the sentence “Followed by, all the full-texts of the included studies were evaluated against the pre-defined inclusion and exclusion criteria.”

Row 129: type of specimen is missing as an extracted data, please add it.

RESULTS

Rows 161-163: Remove the sentence “Further, the studies which doesn’t have sufficient homogenous data or there is an insufficient number of studies to perform meta-analysis were not considered for meta-analysis.” into methodology – study selection subsection

Rows 164-167: The sentence “The reason for the 164 exclusion of studies includes animal studies, case Reports, non-diagnostic studies, 165 treatments of COVID-19, guidelines, protocols, non-English, not RT-PCR techniques, 166 pediatric populations, qualitative research, and reviews.” Also needs to be in methodology – within inclusion and exclusion criteria.

Rows 179-180: Please report the number of studies they were evaluated in.

Table 1.

-        Please sort study design in the 2nd column as follows: prospective (prospective cohorts), retrospective (case-control and retrospective cohorts), and cross-sectional

-        Report data in the 4th column within table 1 in a proper way: Total number of samples/participants with the types of specimen also. This goes for all included studies.

Table 2. I suggest to put this table in a Supplement material

Rows 214-227: These sentences “There 214 was a significantly high level of heterogeneity (I2: 100%; P<0.00001), hence the random effect model was used [Figure 2A].”, “There was a significantly high level of heterogeneity (I2: 100%; 218 P<0.00001), hence the random effect model was used [Figure 2B].”, “There was a 222 significantly high level of heterogeneity (I2: 100%; P<0.00001), hence the random effect 223 model was used [Figure 2C].”, “There was a significantly high level of heterogeneity (I2: 100%; 226 P<0.00001), hence the random effect model was used [Figure 2C].”, “There was 230 a significantly high level of heterogeneity (I2: 98%; P<0.00001), hence the random effect 231 model was used [Figure 2D]”, etc. are unnecessary. They refer to methodology. Delete them from results section.

Subsection 3.12: It is needed to perform Funnel plots for each of the outcomes. Please add those diagrams to Supplementary material S2.

For further work:

1) Please try to do subgroup analysis for diagnostic accuracy parameters (sensitivity, specificity, PPV, NPV, Kappa) of RT-PCR in SARS-CoV2 positive and negative respondents for every specimen separately

2) Please try to do subgroup analysis for diagnostic accuracy parameters (sensitivity, specificity, PPV, NPV, Kappa) of RT-PCR regarding different severity of COVID-19 disease for every specimen separately

3) ) Please try to do subgroup analysis for diagnostic accuracy parameters (sensitivity, specificity, PPV, NPV, Kappa) of RT-PCR regarding different institutions (summarize them into clinics and others) for every specimen separately

4) Check english grammar and language style and spelling. For example Line 70. Lines 374-350, 394-5 over and over repetition of word "various". Use more potential than present tense: could instead of can.

The quality of english language is bellow the quality of the paper. There are grammar and style issues. Some words are repeating too much, even 2-3 times in one sentence. For example Line 70. Lines 374-350, 394-5 over and over repetition of word "various". Use more potential than present tense: could instead of can.

Author Response

(The authors gave the same response as above.)

Round 2

Reviewer 1 Report

Pls see further comments in the attached manuscript on data presentation.

Author Response

Dear Editor/Reviewer,

Thank you so much for your useful comments and suggestions on our manuscript entitled “Diagnostic Performance of Various Clinical Specimens for the Detection of Covid-19: A Meta- Analysis of RT-PCR Studies”. We have modified the manuscript accordingly, and detailed corrections are listed point by point:

Sincerely,

Dr. Khalid I. Bzeizi
Department of Liver Transplantation
King Faisal Specialist Hospital and Research Center
Riyadh, Saudi Arabia
E-mail: [email protected]

Reviewer 2 Report

ABSTRACT

Row 30 – You missed to correct the capital letter C into small one in the sentence: “The Sensitivity and specificity of samples such as 27 nasopharyngeal swabs, saliva, Combined nasopharyngeal/oropharyngeal, respiratory, sputum, 28 broncho aspirate, throat swab…” Please do it.

INTRODUCTION-DISCUSSION

Authors have whole paragraphs repeating. All paragraphs begin with the same word. As in the previous round of reviews, there are extensive repeating of words in adjacent sentences like "various."

MATERIALS AND METHODS

Rows 108: remove underline for Embase

RESULTS

Rows 158: Add all reasons for exclusion (animal studies, case reports, non-diagnostic studies, treatments of COVID-19, guidelines, protocols, non-English, not RT-PCR techniques, pediatric populations, qualitative research, and reviews) just after Following the exclusion of 1863 articles for various reasons…

Table 1.

It is known that observational studies can be prospective (prospective cohort study) and retrospective (retrospective cohort study, case-control). Also, cross sectional studies are specific type. At last, RCTs are experimental, not observational.

So, we kindly recommend to report study design as it is reported in the original article but if it does not suit to any of standard study designs than state “Not clear”

The English language needs to be extensively corrected.

Authors have whole paragraphs repeating. All paragraphs begin with the same word. As in the previous round of reviews, there are extensive repeating of words in adjacent sentences like "various."

Author Response

Dear Editor/Reviewer,

Thank you so much for your useful comments and suggestions on our manuscript entitled “Diagnostic Performance of Various Clinical Specimens for the Detection of Covid-19: A Meta-Analysis of RT-PCR Studies”. We have modified the manuscript accordingly, and detailed corrections are listed  point by point:

Sincerely,

Dr. Khalid I. Bzeizi
Department of Liver Transplantation
King Faisal Specialist Hospital and Research Center
Riyadh, Saudi Arabia
E-mail: [email protected]
